# Primary health care during the COVID-19 pandemic: A qualitative exploration of the challenges and changes in practice experienced by GPs and GP trainees

Minka Grut[1]*, Gilles de Wildt[1], Joanne Clarke[2], Sheila Greenfield[2], Alice Russell[1]

1 Birmingham Medical School, College of Medical and Dental Sciences, University of Birmingham, Edgbaston, Birmingham, United Kingdom, 2 Institute of Applied Health Research, College of Medical and Dental Sciences, University of Birmingham, Edgbaston, Birmingham, United Kingdom

☉ These authors contributed equally to this work.
* minkagrut@gmail.com

**Data Availability Statement:** All relevant data are within the paper and its Supporting Information files. Results of data analysis - the coding

## Abstract

### Background

The COVID-19 pandemic has rapidly changed general practice in the UK. Research is required to understand how General Practitioners (GPs) and GP trainees adjusted to these changes, so that beneficial changes might be sustained, and Primary Health Care (PHC) can be prepared for future challenges. This study explored the experiences and perspectives of GP and GP trainees during the pandemic.

### Methods

Remote, semi-structured interviews (n = 21) were conducted with GPs (n = 11) and GP trainees (n = 10), recruited from across the UK using convenience and purposive sampling. Interviews were audio-recorded and transcribed verbatim. Interview data were analysed with an inductive thematic approach.

### Results

Five overarching themes were identified: (1) *'Thrown in at the deep end'*; (2) *Telemedicine: 'it needs to be a happy balance'*; (3) *Delayed referrals and 'holding' patients*; (4) *The Covid Cohort–training in Covid*; (5) *Suggestions and lessons for the future of general practice'*. GPs reported a turbulent and uncertain time of major changes to PHC. They described the benefits of technology in general medicine, particularly telemedicine, when used in a balanced manner, highlighting the need for accompanying teaching and guidelines, and the importance of patient preferences. Key tools to help GPs manage patients with delayed referrals to Secondary Care were also identified.

framework and results of thematic analysis and theory-driven coding - can be provided on reasonable request and on agreement with the authors, to ensure that ethical and legal requirements are upheld.

**Funding:** This work was supported by the University of Birmingham. Award/grant number is not applicable. The funders had no role in the study design; in the collection, analysis and interpretation of the data; in the writing of the report; and in the decision to submit the paper for publication.

**Competing interests:** The authors have declared that no competing interests exist.

## Conclusion

Several key changes to general practice occurred as a result of the COVID-19 pandemic, including a rapid uptake of telemedicine. The pandemic exposed the strengths and limitations of normal general practice and highlighted the importance of workplace camaraderie. These findings contribute to the evidence base used to adapt PHC infrastructures as we emerge from the pandemic.

## Introduction

Healthcare systems worldwide have seen major disruptions caused by the SARS-CoV-2 (COVID-19) pandemic [1–4]. The UK has been severely affected: since its first registered case on 29 January 2020 [5], there have been over 9 million cases, resulting in over 140,632 deaths, as of 1 November 2021 [6]. UK government policies and guidance during the pandemic have been criticised by the public and NHS workforce as unclear and quickly changing [7]. Early concerns cited by Health Care Professionals (HCPs) included limited access to Personal Protective Equipment (PPE), delayed publication of public health policies, and failure to support staff with appropriate guidance [8–12].

Healthcare is provided by the NHS through primary and secondary care (SC) services. GPs are the first point of day-to-day healthcare and strive to support the physical and mental health of patients within the community. Where necessary, they are responsible for referring patients to the secondary sector–to hospitals or specialist care. The conventional GP training pathway in the UK is two years foundation training, followed by a three-year GP Specialty Training (GPST) programme. In this study, 'GP trainee' refers to a doctor in this three-year programme. This cohort was included because their responsibilities closely mirror that of trained GPs and are therefore relevant to the overall GP experience. The attitudes and lessons that they take forwards from the pandemic, and the extent to which their training has been affected, are important to fully comprehend so we can know how best to support the new cohort of GPs.

Telemedicine is the practice of delivering healthcare through technology, including telephone and video consulting, and utilising Internet resources. Significant changes to general medicine included a suspension of most non-essential services and a switch to the use of telehealth [13] to reduce face-to-face contact. GPs moved almost entirely online and onto the telephone [14]. Before the pandemic, over 70% of GP consultations were carried out face-to-face. Within weeks this figure had dropped to 23% [15], a change facilitated by doctor-led patient triaging prior to e-consultations, and tools such as AccuRx which is now used by over 80% of UK practices [16]. It integrates with practices' Electronic Medical Records (EMR), allowing patients to communicate with GPs via text messages and video calls, and exchange photos and documents, often negating the need for in-person consultations [16, 17]. The NHS Long Term Plan, a new service model released in January 2019 to be adopted over the next five to ten years, includes the aim of providing every patient with the option of a digital consultation [18]. The pandemic has rapidly accelerated this timeline [15].

General practice has ensured continued healthcare access for communities social distancing and shielding–taking precautionary distancing measures to protect, or 'shield', people who are more vulnerable to COVID-19). In places, this has utilised close collaboration with social workers [19]. GPs were encouraged to limit referrals to create hospital space for COVID-19 patients, creating a routine referral rate '74% lower between the weeks commencing 15 March–21 June 2020 when compared with the same weeks in 2019' [13]. COVID-19 has therefore prevented

potentially necessary hospital visits and created severe delays for those referred to Secondary Care (SC) [13, 20, 21]. Immediate consequences such as a nationwide drop-off in urgent cancer referrals, accompanied by a rise in mental illness, have been observed [17, 21, 22]. To manage this, Advice and Guidance–an electronic messaging service that 'allows a clinician (often in primary care) to seek advice from another (usually a specialist) prior to or instead of referral' [23]–has been an important tool for GPs to access specialist advice on how best to treat and investigate patients in the community, instead of or whilst waiting for a referral. Other models of collaboration have been trialled during the pandemic: for example, Northwest London local PHC teams have monitored COVID-19 patients in the community and communicated concerns or need for escalation to SC [24]. Such programs have the potential to alleviate hospital bed pressures and take advantage of the breadth of knowledge and unique doctor-patient relationships in PHC. PHC teams have also helped implement the nationwide vaccination scheme [25, 26].

Such changes have required those delivering primary care services to rapidly adapt to and adopt new ways of working. A qualitative approach is needed to hear GP and GP trainee perspectives of how the pandemic has challenged and changed general practice. A literature review of published peer-reviewed and grey literature revealed little qualitative research exploring UK GP experiences of the pandemic as of 17 November 2021, except for a mixed-methods study exploring remote consulting in PHC during the pandemic. This observed a successful uptake of telemedicine in 90% of 21 practices but highlighted key risks and limitations including technology difficulties and the fear of missing clinical signs [27]. It was possible to find qualitative studies specific to video-consulting in the UK, however these were discouraging and showed the need for feedback on general tele-consulting [28]. In a qualitative study of 132 GPs in Belgium, Verhoeven *et. al* showed GP concerns including those regarding the continuity of care for chronic patients [29]. This study is also informed by a survey-based cross-sectional study of UK GPs during the pandemic [30], and other explorations of the use of remote consulting in PHC [27, 31]. Outside of qualitative literature, infographic-based data was used by Clarke et al. to establish that the use of remote consulting prior to the pandemic aided GP practices during the pandemic [32]. Since this research was conducted, qualitative studies in PHC have been published in Greece, Saudi Arabia, and in a pan-European study which included England [33–35].

This study aimed to broaden the evidence base in qualitative research, using in-depth interviews to explore the experiences and perspectives of GP and GP trainees during the pandemic. Exploring the lessons from this time will help to anticipate and prepare for future health crises, and identifying any positive changes might be utilised to reimagine the way in which day-to-day GP care is delivered in the future.

## Methods

### Design

A qualitative, exploratory study using semi-structured remote interviews with participant-led conversation [36]. With limited existing research into this area, participants' data guided thematic analysis using an inductive approach [37].

### Setting

NHS-based primary care across England, Wales, and Scotland.

### Participants and recruitment

The target population was GP and GP trainees across the UK who worked before and during the pandemic. A study advertisement (S4 Appendix) was distributed through social media

(Twitter and Facebook), personal networks, and snowball sampling [38]. Interested respondents emailed MG, who then supplied a Digital Information Pack (DIP), to be read through in the interviews (Participant Information Sheet, Demographic Questionnaire, and Consent Form). In the interest of time, convenience sampling was initially employed [39]: those that were given a DIP and maintained their interest (100%) were subsequently recruited. However, social media advertisements quickly gained traction, with over 100 GPs expressing interest within days. Participants were not required to supply any demographic information before the interviews, as the Demographic Questionnaire was conducted verbally at the beginning of interviews (data presented in Tables 2 and 3). Despite this, some participants introduced themselves by volunteering demographic information when expressing interest. The sampling method was therefore adapted to capitalise on the strong response and additional data, with participants sampled purposively [40], using their volunteered information, to achieve a cohort varied by the deprivation level and setting (rural or urban) of their practice, and their professional role. Again, those sampled received a DIP and were asked to confirm their interest prior to setting up an interview. Recruitment occurred alongside data collection and analysis until theoretical data saturation was reached at 21 interviews (11 GPs, 10 GP trainees). This was determined through discussions within the coding team. Two further interviews were conducted at this point to confirm that no new frequently appearing codes were arising [41–43]. The participants and researcher had no relationship prior to study commencement. Every person sent a DIP chose to participate. Participants each received a £30 voucher as a token of thanks.

## Data collection

Interviews were conducted MG (female medical student with training in qualitative methodologies) in February/March 2021 via Zoom video call (n = 20) or telephone (n = 1) [44] at the participant's preference, lasting an average of 36 minutes (range 24 to 58 minutes). Calls were used to adapt to Covid-19 restrictions [44, 45]. Interviews were structured by a Topic Guide (S1 Appendix), refined following a pilot interview, and adjusted throughout data collection to explore emerging topics (Table 1). Pilot data was not included in analysis. To begin interviews, participants were informed of the researcher's background, the purpose of the study, and what to expect from the interview. They were then able to ask questions. Verbal consent was taken before the interview. The audio recordings of these were stored on an encrypted memory stick. The demographic questionnaire and interview followed, the audio-recordings of which were stored on a second encrypted memory stick. Each interview concluded with a brief discussion to ensure all issues salient to the participant had been raised. This was necessary to mitigate the study's time constraints, which did not allow for repeat interviews or participant feedback to transcripts and results before the dissemination of results. The audios were recorded using an in-built Zoom recording function, transferred to their respective memory sticks, and deleted from the Zoom server. Field notes were made by the interviewer after each interview [46].

## Analysis

MG manually transcribed interviews as an orthographic record of non-verbal and verbal utterances for data accuracy [47], redacting any potentially identifiable information.

GP and GP trainee interview data were thematically analysed together, employing Braun & Clarke's 6-step recursive analytical approach [47]. This facilitated an in-depth study of participants' perceptions and experiences [47] and organisation of the salient data by codes and themes. To improve the reliability of the coding framework, three transcripts were triple coded

**Table 1. Topic guide overview.**

*Professional Experience of the Pandemic*

**1. Can you tell me about your professional experience of the pandemic?**
  *How prepared did you feel for the pandemic as a GP?*
  *How well informed did you feel your patients were about the pandemic?*
  *How did you feel about making decisions with the guidance you had?*
**2. In what ways did common practice change for you?**
  *Tell me about any changes to clinical guidelines, and how effective you found these adjusting to COVID-19?*
  *In your experience, how have GPs been utilised, and have they had to take on any new roles?*
  *What changes, if any, has the pandemic had on interactions between GP and hospital staff?*
  *Has COVID-19 changed your relationship with your colleagues or your patients?*

*Government Response to the Pandemic*

**3. What is your opinion of the government's response to the pandemic?**
  *How effective have they been in controlling the pandemic?*
  *What is your opinion of the public health messages and policies?*

*Personal Experience of the Pandemic*

**4. Has COVID-19 had any impact on you?**
  *Are you or your family in an at-risk group for COVID-19?*
  *What effect has this had for you as a GP?*
  *What protective measures have you taken for yourself?*
  *If COVID-19 has had any mental or physical impacts on you, do you feel this has influenced your practice, and how?*

*Future of General Practice*

**5. Are there any changes which you think should be carried on into the future? Why and how?**
  *Are there any that should not be?*
  *Can you describe the future of general practice in recovering from this pandemic?*
  *What do you think we can learn from the pandemic?*

*Trainee-Specific Questions*

**6. How do you think the pandemic has influenced your training?**
  *Has it had any effect on your views of general practice or informed your specialty choice?*

independently by MG, JC, and AR, who then compared codes. This use of triangulation [36] revealed very similar interpretations of the data, and where these differed, it proved conducive to a well-refined coding framework of five overarching categories. Coding was informed by Saldaña's two-cycle approach [48], to ensure thorough data analysis. Using NVivo 12, MG used the coding framework to code the remaining 18 transcripts. Constant comparison [49] made between interviews allowed new codes to be drawn from the data and applied to the framework. Deviant case analysis was used to ensure the framework was flexible and not limited by analytical bias [50]. Categories were developed into themes.

To encourage external validity and ensure the most relevant and insightful data were presented, preliminary themes were discussed with the research team. To improve credibility of the results, transparency and reflexivity were maintained throughout the study [51].

The 'Consolidated criteria for reporting qualitative studies (COREQ): 32-item checklist' was used to report the study (S2 Appendix) [52].

## Ethical considerations

This study was approved by the BMedSc Population Sciences and Humanities Internal Research Ethics Committee, University of Birmingham (Reference Number: IREC2020/1761957). The Committee approved the contents of the study advertisement. All participants gave verbal consent before commencement of interviews, and data protection and confidentiality protocols were followed.

**Table 2. Distribution of demographic variables within sample (N = 21).**

| Characteristic | | Category | | % | n (N = 21) |
|---|---|---|---|---|---|
| Professional Role | | GP | Partner | 4.8 | 1 |
| | | | Partner + Locum | 4.8 | 1 |
| | | | Locum | 9.5 | 2 |
| | | | Locum + Salaried | 4.8 | 1 |
| | | | Salaried | 28.6 | 6 |
| | | GP Trainee | | 47.6 | 10 |
| Age | | <30 | | 38.1 | 8 |
| | | 30–39 | | 47.6 | 10 |
| | | 40–49 | | 4.8 | 1 |
| | | 50–59 | | 4.8 | 1 |
| | | >60 | | 4.8 | 1 |
| Gender | | Female | | 57.1 | 12 |
| | | Male | | 38.1 | 8 |
| | | Other | | 4.8 | 1 |
| Ethnicity | | White | | 47.6 | 10 |
| | | Asian or Asian British | | 38.1 | 8 |
| | | Black, African, Caribbean, or Black British | | 9.5 | 2 |
| | | Mixed or multiple ethnic groups | | 4.8 | 1 |
| Total years | Worked in general practice (GP) (N = 11) | Median | | 9.32 (range 1–28) | |
| | Year of training (GPTr) (N = 10) | ST1 | | 10 | 1 (N = 10) |
| | | ST2 | | 10 | 1 (N = 10) |
| | | ST3 | | 70 | 7 (N = 10) |
| | | ACF4 | | 10 | 1 (N = 10) |
| Practice Setting | | Urban | | 47.6 | 10 |
| | | Suburban | | 42.9 | 9 |
| | | Rural | | 9.5 | 2 |

Partner GP = a self-employed GP, responsible for running their own practice

Locum GP = a fully-qualified GP that fills the vacancy of a permanent staff (for example, who is away on sick leave)

Salaried GP = a GP who receives a basic monthly salary for their permanent role within a practice

STx = (GP) Speciality Training x (where x = year of training) [53]

ACFx = (GP) Academic Clinical Fellowship x (where x = year of training)

## Results

Eleven GPs and 10 GP trainees were recruited from rural, suburban, and urban practices across England, Wales, and Scotland. Many participants described their practices as serving ethnically and socioeconomically diverse populations. Participants had a mean age of 35 years (range 28–63 years). Twelve participants described themselves as female, eight as male, and one as other. The distribution of demographic variables is given in Table 2, and a more detailed demographic for each participant is given in Table 3.

Thematic analysis identified five core themes (Table 4). Themes are supported with participant quotes presented in clean verbatim style. 'GP' is in reference to both fully qualified and training GPs, unless otherwise specified.

### 1. 'Thrown in at the deep end' (GP10)

**An overnight change.** For the majority of participants, the UK lockdown [54] prompted an overnight shift to remote consultations. Participants found themselves having to social distance from others, and utilise teleconsulting with little or no prior experience.

**Table 3. A description of each participant's (N = 21) demographic and practice characteristics.**

| Participant ID | Age Group | Gender | Ethnicity | Practice Setting |
|---|---|---|---|---|
| GP1 | 50–54 | Female | Asian or Asian British | Urban |
| GP2 | 30–34 | Male | Asian or Asian British | Urban |
| GP3 | 30–34 | Female | Asian or Asian British | Urban |
| GP4 | 25–29 | Female | White | Urban |
| GP5 | 35–39 | Female | White | Suburban |
| GP6 | 30–34 | Female | Asian or Asian British | Suburban |
| GP7 | 35–39 | Male | Asian or Asian British | Suburban |
| GP8 | 35–39 | Male | Asian or Asian British | Urban |
| GP9 | 35–39 | Male | White | Urban |
| GP10 | 60–64 | Male | White | Rural |
| GP11 | 40–44 | Female | Asian or Asian British | Suburban |
| GPTr1 | 30–34 | Other | White | Suburban |
| GPTr2 | 25–29 | Male | Asian or Asian British | Urban |
| GPTr3 | 25–29 | Female | Black, African, Caribbean, or Black British | Rural |
| GPTr4 | 25–29 | Female | Black, African, Caribbean, or Black British | Urban |
| GPTr5 | 25–29 | Male | White | Suburban |
| GPTr6 | 35–39 | Female | White | Suburban |
| GPTr7 | 25–29 | Female | White | Urban |
| GPTr8 | 25–29 | Female | Mixed or multiple ethnic groups | Suburban |
| GPTr9 | 30–34 | Female | White | Suburban |
| GPTr10 | 25–29 | Male | White | Urban |

GP = General Practitioner; GPTr = General Practitioner Trainee

*'It's changed the nature of the practice. Pretty much overnight. So, we went from 100% face-to-face to 99% telephone and video consultations.' (GP10)*

*'Generally it was quite a frightening time for everyone, both doctors and patients included.' (GP2)*

*'I found it quite isolating' (GP8)*

**Lack of support.** Over half of participants reported insufficient guidance from governing bodies for these changes. Often, practice managers autonomously decided to close their practice doors and move to remote healthcare delivery.

*'I don't think we've ever had any decent guidance. . . we were kind of left to our own devices. . . it's just 'follow the news, make it up as you go along'.' (GP9)*

Any guidance provided was described as quickly changing, '*patchy*' and '*a bit vague*' (GP4), or inappropriately delivered.

*'Clinical leadership hasn't been great in terms of compassion. . . lots of people have found the emails sent from the clinical director quite affronting at times, particularly when colleagues were concerned about PPE and things like that. . . there was one where the email was quite sarcastic.' (GPTr6)*

A few GPs described the responsibility of clarifying government policies for patients.

Table 4. **Themes and subthemes.** As identified in thematic analysis.

| THEME | SUBTHEME |
|---|---|
| 1. 'THROWN IN AT THE DEEP END' | a. An overnight change<br>b. Lack of support<br>c. Filling the gaps<br>d. Camaraderie |
| 2. TELEMEDICINE: 'IT NEEDS TO BE A HAPPY BALANCE' | a. Negative impressions<br>b. Positive impressions<br>c. The benefits of AccuRx |
| 3. DELAYED REFERRALS AND 'HOLDING' PATIENTS | a. Impacts of delayed referrals<br>b. Advice and Guidance |
| 4. THE COVID COHORT–TRAINING IN COVID | a. Changes to teaching<br>b. Support |
| 5. TIME FOR CHANGE | a. Recovering from the pandemic<br>b. Four suggestions for the future |

*'The shielding guidance is not very clear, some people felt they should be shielding but they didn't receive the letter, but some people will receive the letter. . . we aren't able to figure out why' (GP11)*

Multiple participants perceived an increased responsibility and decreased consulting confidence when managing COVID-19 cases and risk stratifying non-COVID patients by telephone. This was compounded by failed phone-triaging guidance, such as the Roth score index, used to assess breathlessness [55].

*'In the first wave there was this big thing about using the Roth score. . . after about two weeks there was this big thing about the Roth score's not safe. . . and everyone thinking oh my God, I spoke to so many patients last week and thought they were fine. . . that was quite difficult.' (GPTr10)*

*'I was speaking to quite a few patients on the phone who had Covid. . . and it was quite scary. . . knowing that there wasn't too much you could do other than call an ambulance and. . . the standard guidance.' (GP4)*

**Filling the gaps.**   GPs often described searching for information to help them safely conduct telephone consultations. Sources of guidance included '*collaborative chat on social media*' (GP3), practice networks, online GP support groups, and medical journals.

*'There were some really good articles, mostly from the BMJ. . . that helped me know how to triage patients with Covid, and I used those as a. . . safety net'. (GP5)*

*'. . .various sources. . . emails being disseminated through the primary care network, NHS England, from the Government, Public Health, just my own research, and things that people had collated together in various resources for GPs. So, things like GP notebook, NV Medical Reviews, Medwell Reviews. . . even things like Facebook!' (GP6)*

**Camaraderie.**   Given the uncertainty shared amongst practice staff, participants valued camaraderie and the reassurance they felt navigating the pandemic '*as a team, supporting each other*' (GPTr2).

## 2. Telemedicine: 'It needs to be a happy balance' (GP11)

Despite the initial uncertainty from the switch to telemedicine, many GPs perceived the change as part of an inevitable positive shift towards technology, accelerated by the pandemic.

> *'I think the changes that we had to make overnight would- should have happened a while ago, actually, but my practice is in the dark ages a little bit!. . . suddenly we've been pushed out of our comfort zone and find that things work better.' (GP6)*

Participants had both negative and positive impressions of this shift, although the majority agreed that, if used correctly, teleconsulting has a place in future PHC.

**Negative impressions of telemedicine.** Participants expressed concerns regarding their ability to develop strong and reassuring doctor-patient relationships on the phone. Trainees described the difficulties they faced trying to '*build up the trust. . . as a new doctor in that surgery*' (GPTr6), reporting that they instead felt like a faceless '*call centre*' or '*convenience store of general practice' (GPTr7)* for their patients.

> *'It's been tougher than I thought it would be. . . you miss that face-to-face contact. . . it's very difficult to make a sustainable GP relationship with a person you don't meet face-to-face' (GP8)*

GPs also found many patients to be dissatisfied without face-to-face consultations, perceiving telephone calls to be '*somehow substandard' (GP2)*, or incomplete.

> *'A lot of our patients like that face-to-face interaction, and sometimes it can be difficult explaining to them that we can sort out this problem from the history. . . they find that difficult. . . (GP4)*

The most frequently cited difficulty of telemedicine was safely risk-stratifying patients over the phone, due to loss of visual cues and managing '*difficult historians' (GPTr7)*, particularly those from the '*elderly, frail, comorbid population' (GPTr8))*. '*Good safety netting' (GPTr9)* was emphasised as a means of adapting to such barriers.

> *'This was tough, because we'd gone from seeing people face-to-face, examining them, and having [observations], to having to. . . make an assessment based on how they sound on the phone.' (GP6)*

When significant communication barriers threatened participants' ability to effectively risk-stratify, many described a low threshold for converting the consultation to a face-to-face, particularly for more vulnerable patients.

> *'If I'm not really winning with someone over the phone, I just bring them down, and that's obviously a risk-benefit weigh up every time. . . the worry is the. . . cues that you haven't picked up on. . .' (GPTr8)*

Many observed that socially disadvantaged patients faced additional barriers of being less likely to have access to a mobile phone or private space from which to call their doctor, exacerbating socioeconomic inequalities.

*'Lots of our patients don't have mobile phones, if they do have mobile phones are on a cheap contract where they don't get lots of data. . . or it costs you to access a voice message' (GP5)*

Most GPs described telemedicine as giving patients '*rapid access to diagnostic care' (GP5)*. Whilst this reduced waiting times, it also increased their workload and complicated risk stratification further by requiring participants to assess a patient '*very early in the path of physiology' (GP5)*.

**Positive impressions of telemedicine.** Most participants described the shift to telephone consulting as more accessible for certain patients, particularly the younger population.

*'A lot of people, especially younger people who generally are at work. . . can just have a phone call in the middle of the day. They don't have to take time off work, come physically face-to-face to see me, so I think that's a real positive.' (GPTr1)*

This was a mutually beneficial outcome of teleconsulting—the majority of participants described telemedicine as '*far more time-efficient' (GP2)* than full days of face-to-face appointments. By saving time on consultations '*that could be easily dealt with over the phone' (GPTr8)*, they were able to allocate time more efficiently.

Not being in front of their patients made some more comfortable asking colleagues for advice or utilising resources from the National Institute for Health and Care Excellence (NICE), including Clinical Knowledge Summaries (CKS) [56] and the British National Formulary [57]. This was reassuring, particularly for trainees.

*'There can be a lot of pressure to just know things. . . to deliver the best, most evidence-based care to our patients, we should be looking at that evidence base. . . It is even easier to do that over the phone. Because I can be talking to them, and I can have NICE CKS up. . .. I find that reassuring.' (GP5)*

**The benefits of AccuRx.** AccuRx Desktop was frequently cited as an example of positive innovation, described as '*brilliant' (GPTr5)* and '*an incredible tool' (GP1)*. GPs used it to send fit notes (official statements addressed to a patients' workplace during sickness absence), referral letters, imaging requests and blood forms.

## 3. Delayed referrals and 'holding' patients (GPTr10)

**Impacts of delayed referrals.** As a result of delayed referrals to SC, some participants reported significant changes to their responsibilities. Delays were most often to usual referrals, but occasionally two-week referrals had also been extended.

*'I've had letters back from secondary care saying. . . patients who have been on a 2-Week-Wait or a Rapid Access Cancer Pathway, you should not tell them they'll be seen soon because they won't be. Which is really hard, to speak to cancer patients. . .' (GPTr6)*

Multiple examples were given of patients whose referrals had been delayed. These delays were often described as damaging to patients' mental and physical health, with consequences including dependence on '*really strong painkillers' (GP3)*.

*'One patient. . . has a really high marker for inflammatory bowel disease. . . waiting for six months. . . lost their job. . . is now almost suicidal. . . We find that patients really do just*

*deteriorate over the months. . . trying to cope with these waiting lists and the uncertainty.'
(GPTr6)*

This was often *'difficult and stressful' (GPTr10)* for participants, many of whom reported
they were *'taking more risk and avoiding referral' (GP9)*. Some became the *'point of contact for
[patients] venting' (GPTr5)*.

**Advice and guidance.**   Participants frequently reported Advice and Guidance as a key tool
in managing delayed referrals, allowing them to deliver the best care to patients in the PHC
setting whilst they waited to be seen by SC.

*'I can prevent a referral that perhaps doesn't need to be a referral and would be a delayed
wait, or admission to A&E. . . feedback from specialists has been really useful and has enabled
me to just manage them in general practice' (GPTr4)*

Some described themselves to be *'quite excited to be taking on more responsibilities from the
hospitals' (GP7)*, whilst others were unhappy with the additional burdens. There were accounts
of miscommunication with secondary care, for example regarding who was responsible for
ordering tests. One GP reported that not referring their patients had left them *'feeling. . .
completely useless and everything being futile' (GP9)*.

*'We've had to take a lot of burden away from secondary care. . . The complexity of patients'
care is massively rebounding on us. . . we're then having to make decisions and do things that
might be further out of our comfort zone.' (GPTr10)*

## 4. The Covid Cohort–training in Covid

**Changes to teaching.**   This theme drew solely from GP trainee responses. Most trainees
mentioned the change of their exam format, from the Clinical Skills Assessment (CSA) to the
Recorded Consultation Assessment (RCA) [58], as a notable impact COVID-19 had had on
their training. Whilst some described the RCA as a *'tricky exam' (GPTr7)*, multiple participants
felt that *'the adaptions to the exam [had] been pretty good' (GPTr8)*.

*'My CSA got cancelled. . . I had to rethink it all and prep for the RCA which was really
challenging. . . not knowing what was happening and whether I was going to finish training or
not was. . . one of the hardest things for me really to deal with psychologically through the pan-
demic.' (GP4)*

Trainees experienced a total drop-off in teaching at the start of the pandemic followed by a
change to virtual learning, creating concerns about the loss of teaching time and the lack of
exposure to non-pandemic GP care.

*'I'm pretty sure I'm going to qualify in August having never done a full face-to-face GP clinic.
Which I find kind of horrifying.' (GPTr7)*

**Reduced support.**   Some described the experience of GP training during the pandemic as
*'overwhelming' (GPTr10)*.

*'Having to do telephone appointments for patients . . . it's very difficult to make a decision. . .
as a trainee you have more pressures on you. . . I definitely have felt burned out.' (GPTr10)*

They often described feeling *'lonely' (GPTr2)* following the loss of networking with other trainees, whilst having to distance from their usual support networks outside of work.

*'It's been really difficult in terms of not being able to socialise with people, and I think that's part of our GP training, just to have a community of support. . .' (GPTr4)*

*'It's had quite a big impact. . . I actually have a niece who's 18 months old and I've only seen her once. . . so that's been really hard. . . anxiety. . . I haven't touched my parents in a year. . . I haven't seen a lot of my non-medic friends for almost a year. . . one of my best friends. . . got really, really depressed, and I couldn't see her.' (GPTr6)*

GP Trainees had variable experiences of the guidance they had received. Often, it seemed to depend on how well-supported they had been by their practice, although examples of support from other governing bodies were given, for example, *'e-learning modules' (GPTr2)* from the Royal College of General Practitioners (RCGP).

*'We frequently get invites to meetings with the chief executive and the chair of the Junior Doctor committee. . . where we can raise concerns, ask questions, learn about the stats. . . frequent emotional support.' (GPTr5)*

For those on their hospital placement at the start of the pandemic, the extension of their four-month specialty rotation to eight months was not well-received.

*'I ended up having to stay for eight months. . . on an acute Covid ward, and it was horrible. . . very disheartening, you already felt like you weren't looked after, and then you're forced to stay somewhere you don't want to be anymore.' (GPTr3)*

## 5. Time for change

After a tumultuous time for general practice, participants were keen to reflect on the ways in which their work has changed. The next steps will lie in recognising this period of great uncertainty for GPs and learning from how they adapted as we move forwards.

**Future challenges.** Participants voiced concerns regarding the future healthcare problems general practice would be responsible for, caused directly and indirectly by the pandemic. These concerns included missed chronic care reviews and screening, and mismanaged acute presentations.

*'There's going to be this huge backlog of secondary care stuff and this worry about missed diagnosis . . . I suspect there'll be an ongoing burden with mental health because there's going to be big unemployment, patients that will have unfortunately lost relatives and loved ones. . . it's likely to have complicated life for a lot of people.' (GPTr10)*

**Suggestions for the future.** GPs provided suggestions for how general practice should change, incorporating useful changes from the pandemic to face the challenges discussed above.

1. The search for a telemedicine hybrid with effective triaging
   GPs agreed that *'remote working and telemedicine. . . [is] here to stay' (GP7)*, but all believed that a balance between remote and face-to-face consulting is needed.

*'We need to accept and embrace digital changes and remote changes to patient care. . . but we've also got to learn how we merge that with the old way of working. . . to find a hybrid model.' (GP8)*

Triaging, used to determine which patients needed to be seen in person, led some GPs to take a first history of patients over the phone and a second history once face-to-face. This could leave them feeling they were *'wasting [their] time double consulting people' (GPTr7)*. Suggestions for a *'more streamlined' (GPTr10)* service were given.

*'There tends to be a lot of duplication if you speak to them on the phone and then you say, OK you need to now come in and I need to examine you, and then you end up going through the whole thing again.' (GPTr10)*

2. Faster specialist advice for GPs

Five participants described positive experiences of Advice and Guidance and made suggestions for future use, to improve its accessibility and clarify its role in general practice.

*'Another thing that I think could be a little bit more accessible is Advice and Guidance pathways as well? So perhaps having a specialist on-call just for GPs. . . I think more of that. . . would prevent a lot of referrals and a lot of sending patients to A&E.' (GPTr4)*

3. Public education

Participants frequently described a perception among patients *'that general practices were closed' (GP10)*, causing patient anxiety and delayed presentations, in turn creating difficulties for GPs. It was suggested that this could be countered through public education on accessing GP care, reinforcing practices were open but consulting mainly via telephone and that this was an appropriate model of care.

*'A lot of the fears are about consulting, as in patients coming in. . . GP groups. . . need to push for more health campaigns to say we are still open. . . if you need something just call us, don't sit on it'. (GPTr3)*

4. Professional education, including clinical leadership and guidance

To combat vaccine hesitancy amongst patients, multiple participants recommended GP education, with improved communication between HCPs and healthcare academics, to better translate science into clinical practice.

*'What could be useful. . . if the GPs did have mandatory teaching about the vaccines, the pros and cons, and also a bit of education about vaccine hesitancy. And how to educate and help people overcome that.' (GPTr4)*

*'What you want is a clear pathway, between academia and clinical medicine. . . to communicate. . . their findings. . . One of the reasons why we've had so many myths. . . about Covid. . . is because we haven't been able to communicate the message.' (GP7)*

Regarding clinical leadership, many participants, particularly trainees, felt let down by support provided during the pandemic and believed this might have been prevented with effective leadership skills teaching.

*'I think the clinical leadership hasn't been great in terms of compassion. . . So, I think greater equipping with actual leadership skills. . . would be much better.' (GPTr6)*

## Discussion

This study presents GP and GP trainee experiences of the rapid near-total shift to teleconsulting, delayed and reduced referrals to secondary care, and interrupted training for GP trainees. Participants identified positive and negative features of teleconsulting and stressed the need for a balance between face-to-face and remote consultations. The findings, particularly those regarding a need for COVID-19-relevant training and support with the quick adjustment to telemedicine during the pandemic, align with recent published qualitative studies in Greece and Saudi Arabia, and an exploratory study across eight European countries including England [32–34].

### Support

Almost all participants described the beginning of the pandemic as a tumultuous period. Participants were encouraged to use telemedicine wherever possible yet reported having little-to-no consistent guidance to accompany this. The pathway through which information is officially disseminated to GPs appears convoluted. Official guidance, delivered a few weeks into the pandemic collaboratively and separately by the NHS, the RCGP, the British Medical Association (BMA), and the Care Quality Commission (CQC), was found by participants to be delayed, conflicting, and confusing. This created an initial period of significant uncertainty, leading GPs to seek unofficial sources of information, including online GP support groups and reports by the British Medical Journal (BMJ), for example, Greenhalgh. T's 'Covid-19: a remote assessment in primary care' [59] and 'Video consultations for COVID-19' [60]. Improving GP consulting confidence with strong leadership and guidance promotes the safety of triaging and hence patient care. Participants often found that an incomplete understanding of government guidelines and shielding policies prevented them from reassuringly explaining these to patients.

Lim *et. al* established that a united front in the face of healthcare challenges like COVID-19 was a key factor in the successful Singaporean PHC response [61]. In this study, strong communication channels were key to keeping GPs informed whilst practicing in the pandemic. The importance of workplace camaraderie has also been highlighted in a recent unpublished qualitative study (S3A Appendix). Effective communication is equally vital between PHC and governing bodies, to produce guidance reflective of observations made by frontline workers, that is then appropriately disseminated to the necessary HCPs. This includes communication channels between HCPs and academics, for professional education to effectively combat problems like vaccine hesitancy. An unpublished qualitative study in Ireland reflects the importance of such channels (S3B Appendix).

Strong communication channels between GPs and SC have also become important to general practice. Participants in this study described the additional responsibility and risk of caring for patients for a longer time than usual due to significant waits for specialist care. Managing conversations with frustrated patients whilst navigating new relationships with SC proved particularly difficult. This was partially alleviated by Advice and Guidance, through which participants were able to obtain specialist SC advice and safely care for patients awaiting referral. Going forwards, closer collaboration between primary and secondary care will be paramount to addressing the backlog of referrals [17, 62]. Arguably, GPs will require appropriate funding and support if they are to continue taking on diagnostic and treatment responsibilities from SC.

### Benefits and drawbacks of telemedicine

Participants reported timesaving as a benefit of telemedicine. This was reflected in a cross-sectional survey, assessing GPs' experiences of the pandemic [30]. However, participants expressed concern about their ability to cultivate a strong doctor-patient relationship and to confidently risk-stratify, particularly when a patient presented early in the disease course. Participants often attributed these difficulties to their lack of prior teleconsulting experience, reflected by Clarke et al., who suggested that UK practices that already offered remote consultations (pre-pandemic) were better prepared for its use in the pandemic [32]. This highlights a need for supportive training in any move towards teleconsulting, as encouraged by NHS England [63]. A multi-method evaluation of eConsult (a form of telemedicine piloted in Scotland) suggested that having a practice-based 'eConsult champion' might assist GPs in safely delivering teleconsultations [64]–this finding might be applied to telemedicine use.

Participants made great efforts taken to recognise the experiences of their patients, likely because this directly impacts their own satisfaction with consultations. They reported additional barriers for patients from more deprived socioeconomic backgrounds, and difficulty obtaining histories from elderly patients over the telephone. In this way, telemedicine might compromise proportionate universalism [65] and inflate pre-existing health inequities [27, 66, 67]. This concern is mirrored in existing research exploring telemedicine in PHC [27, 30, 31]. Norman et al. have highlighted this consequence in their qualitative study focussed on COVID-19's effect on PHC in areas of high deprivation [68]. However, in other instances remote consultations were found to improve accessibility, such as for housebound patients or those with children or full-time jobs. A future integrated model of remote and in-person consultations must be able to adapt to patient factors including background, presentation, preferences, and expectations.

### Alternative technology

In contrast to audio calls, participant experiences of video calls were overwhelmingly negative. Whilst their use has been encouraged [69], particularly for dynamic (ie. movement) and visual presentations, participants found that technology barriers like unreliable Wi-Fi rendered video calls a less practical option than telephone calls. Similar negative experiences with video calls were documented in other UK qualitative studies of remote consulting in PHC during the pandemic [27, 28]. In contrast, the messaging feature on AccuRx was popular amongst participants, who used it to contact their patients and SC, likely explaining its uptake in GP practices during the pandemic [16, 27].

### Missed learning opportunities for GP trainees

GP trainees believed their training had been significantly disrupted by the change in exam format, from the CSA to RSA, and decreased formal training. Whilst some adapted comfortably to these changes, others were concerned by the impact on their training, and the future implications for their performance as GPs. Participants who felt supported by their practice were more confident in their abilities and less affected by the loss of trainee networking, again demonstrating the importance of workplace camaraderie. The General Medical Council (GMC) National Training Survey 2020, which covered 38,000 trainers and trainee doctors, found that 69% of trainees felt their training had been disrupted, and 82% agreed that COVID-19 had limited their ability to meet their training requirements. Nearly half of junior doctors experienced burnout, depression, or anxiety, and the majority agreed that loss of regular teaching sessions had eliminated valuable educational and socialising opportunities, and much-needed peer support [70]. These reflect our study's findings. Ongoing initiatives to establish virtual

peer support groups and better advertisement of available wellbeing resources will be important for supporting those who have trained during the pandemic [70].

## Criticism of GPs in UK media and politics

This study is also relevant for the debate on GPs' perceived unwillingness to see patients face-to-face that gathered momentum some months after the interviews were complete in April 2021. Particularly from September 2021, GPs saw a surge of negative media levelled against them, for example the *Telegraph's* (newspaper) headline 'GPs still ignoring orders to allow patients face-to-face appointments' and the *Times'* (newspaper) claim that 'virtual GP visits are 'costing lives'' [71]. The RCGP College Chair Martin Marshall described GPs as 'demoralised and exhausted', in a letter responding to the criticism [72]. In news outlets and online polls, GPs have reported severe pressures in terms of resources and public expectations, with a lack of government and public support [71–76]. This study gives GPs a voice to describe their experience of working on the frontlines during the pandemic, the dissemination of which might be an important step towards remedying GP-public relations. The BMA has reported 16% of 2,050 GP members plan to leave the NHS altogether [74]. Improved support, and better staff protection, will be essential if NHS England are to maintain and grow the GP workforce as planned [77].

## Strengths and limitations

Recruitment yielded a cohort varied by both participant and practice characteristics, with participants assuming a variety of roles within practices and key decision-making groups, producing a rich and novel dataset. A limitation of recruitment was the potential for self-selection bias, with those expressing interest in the study potentially having had a more extreme experience. Furthermore, most participants were recruited via social media, resulting in a relatively young cohort. Findings may therefore not reflect the views of more established GPs; however, they were in line with a recent unpublished study investigating video-consulting, which sampled an older cohort of GPs (S3C Appendix). Attrition bias was avoided by a 100% response rate from the point of first response from participants.

All interviews were conducted remotely facilitating a wide geographical spread of participants which improved the transferability of findings. However, reduced receptivity to non-verbal cues may have compromised contextual data collection [78]. Repeat interviews and member validation were not deemed possible under time constraints. Credibility of the study findings was instead ensured by triple coding and analyst triangulation [79].

## Conclusion

This study highlights the importance of strong communication and support systems within general practice teams during the pandemic. This is critical for GPs taking on more responsibilities from SC and adapting to new technologies. Dealing with the exceptional circumstances of the pandemic also alerted GPs to their need for relevant health communication training and stronger channels of communication from governing bodies. COVID-19 has prompted a rapid integration of telemedicine into standard practice, exposing key strengths and limitations of this model of consulting. This study's findings are supportive of its adoption within a hybrid model of remote and in-person care, which could be made safe and accessible by accounting for patient preferences and needs. Findings might also help prepare for potential future, largescale health and healthcare challenges. Finally, findings on facilitators and barriers may also inform the development of improved strategies and approaches for better recruitment and retention of the GP work force.

## Supporting information

**S1 Appendix. Topic guide.**
(PDF)

**S2 Appendix. COREQ-32 checklist.**
(PDF)

**S3 Appendix. Unpublished references.**
(PDF)

**S4 Appendix. Social media advertisement.**
(PDF)

**S1 Data.**
(ZIP)

## Acknowledgments

Thanks to all the study participants who kindly gave their time to this study.

## Author Contributions

**Conceptualization:** Minka Grut, Gilles de Wildt, Joanne Clarke.

**Formal analysis:** Minka Grut, Gilles de Wildt, Joanne Clarke, Alice Russell.

**Funding acquisition:** Minka Grut.

**Investigation:** Minka Grut.

**Methodology:** Minka Grut.

**Project administration:** Minka Grut, Gilles de Wildt.

**Writing – original draft:** Minka Grut.

**Writing – review & editing:** Minka Grut, Gilles de Wildt, Joanne Clarke, Sheila Greenfield.

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
