## [Decision Letter · Decision Letter 0]

16 May 2022

PONE-D-21-40820Primary Health Care during the COVID-19 pandemic: a qualitative exploration of the challenges and changes in practice experienced by GPs and GP traineesPLOS ONE

Dear Dr. Grut,

Thank you for submitting your manuscript to PLOS ONE. After careful consideration, we feel that it has merit but does not fully meet PLOS ONE’s publication criteria as it currently stands. Therefore, we invite you to submit a revised version of the manuscript that addresses the points raised during the review process. Please see the comments from both the reviewers and address them in the revised manuscript. 

We look forward to receiving your revised manuscript.

Kind regards,

Alok Ranjan

Academic Editor

PLOS ONE

**Journal requirements:**

“This work was supported by the University of Birmingham. Award/grant number

is not applicable. The funders had no role in the study design; in the collection, analysis and interpretation of the data; in the writing of the report; and in the decision to submit the paper for publication.”

“Funding for this research was provided by the University of Birmingham. Funders had no role in study design, data collection and analysis, preparation of manuscript, or the decision to publish.”

“This work was supported by the University of Birmingham. Award/grant number

is not applicable. The funders had no role in the study design; in the collection, analysis and interpretation of the data; in the writing of the report; and in the decision to submit the paper for publication.”

6. Please note that in order to use the direct billing option the corresponding author must be affiliated with the chosen institute. Please either amend your manuscript to change the affiliation or corresponding author, or email us at plosone@plos.org with a request to remove this option.

7. Your ethics statement should only appear in the Methods section of your manuscript. If your ethics statement is written in any section besides the Methods, please move it to the Methods section and delete it from any other section. Please ensure that your ethics statement is included in your manuscript, as the ethics statement entered into the online submission form will not be published alongside your manuscript.

**Reviewers' comments:**

Reviewer's Responses to Questions

**Comments to the Author**

1. Is the manuscript technically sound, and do the data support the conclusions?

Reviewer #1: Partly

Reviewer #2: Partly

2. Has the statistical analysis been performed appropriately and rigorously? 

Reviewer #1: N/A

Reviewer #2: N/A

3. Have the authors made all data underlying the findings in their manuscript fully available?

Reviewer #1: No

Reviewer #2: Yes

4. Is the manuscript presented in an intelligible fashion and written in standard English?

Reviewer #1: Yes

Reviewer #2: Yes

5. Review Comments to the Author

Reviewer #1: Although, the issue is important to explore but I feel there is some scope of improvement in analysis.

1. The author needs to define "general practice". What does general practice entail in context of primary healthcare?

2. Further, the scope of future has been explored only for vaccine, what about other aspects of general practice?

3. What was the rationale of including GP trainees?

4. What was the content of the social advertisement developed for recruitment? Were these also reviewed by ethics committee?

5. Would be useful to lay down the scope of telemedicine in context of different healthcare needs. for example- medical abortion and telemedicine

Reviewer #2: The paper aims to capture the experiences of GP and GP trainees(GPTs) engaged in primary healthcare services in UK during the COVID19 pandemic.

While capturing the experiences and opinions of GPs /GPTs who are in the forefront of delivering primary healthcare during an unforeseen disaster can be a worthwhile research exercise, I feel the paper has many weak areas, which I shall attempt to put forth. I hope the following comments are received as constructive feedback that authors could reflect upon and improve their paper.

METHODOLOGY:

One of the weaknesses I would put forth is in the sampling methodology and size. Though authors have listed as a limitation, the channel of recruitment and the incentives provided, could have resulted in somewhat a homogenous and self-selected group. An average of 36 minutes for interviews seems inadequate to capture broad areas mentioned in interview guidelines and the nuanced aspects given that there was no prior relationship between researcher and participants. The authors could have increased the sample size, though for qualitative study, even lesser would be good, provided there is richness in the data. I would be interested to understand what kind of saturation in data did the researchers reach that they felt this was a good number to stop data collection. For example, even though there were both urban and rural based practitioners (adopted as a sampling strategy to provide diversity), there is no comparison or comment made on these lines in findings/discussion.

The authors claim to have used recursive thematic analytical style. While it is relevant for a research topic of this nature, the thick description that is required for qualitative research is missing in the findings. Central to any qualitative research is the ‘emic’ perspective, but in this paper, there is nothing much discussed about the participants’ contexts except for what they themselves have ‘reported’ as an experience/opinion about the pandemic. For an overwhelming catastrophe of this nature, the emotions of the providers seem to be missing. Probably the researchers were not trained enough to elicit such responses, or these arise due to lack of personalization in telephonic nature of interviews compared to face to face. The interview guidelines also reflect that each topic was not sufficiently exhausted and moved from one broad area to another.

I could get some glimpses of it, such as when the providers mentioned about guilt of delayed referrals, etc. I am also curious to know if the respondents mentioned about their own families as it would have been an important aspect of the GP’s making meaning of the pandemic, themselves, their lives and their patients. Though the guidelines mention a set of questions to explore this, nothing has come out in findings. In addition, I felt the researcher-author’s voice(s) were also missing in the discussion and the paper is almost reduced to a ‘report’ than a research finding.

CONCEPTUALIZATION:

I feel the weakness of the paper is not having a set of well-defined and theoretically backed research questions beyond stating the obvious ‘what the experiences of the GP & GP trainees on are…”. Much of the findings are mere descriptions and even the presentation is based on broad descriptive categories than ‘themes’. For instance the fifth theme is called “suggestions” which by itself cannot be considered as a theme. Similarly the fourth theme on changes to teaching and support could actually be combined with theme 1 as they both speak to need for effective support and guidance. Each theme in the paper received more of a ‘touch and go’ treatment and remain isolated from each other. May be the paper should have focused just on the telemedicine aspect or on the personal/professional dilemmas.

The paper could have elaborated more on how the NHS functions and the role of GPs and general issues faced by the health workforce of that cadre. This would also include on how other building blocks of the health system (financing, leadership & governance, information systems, access to medicines) interact with health workforce to affect health service delivery. In the current version, the paper doesn’t connect the findings to the health systems.

FINDINGS/DISCUSSION:

The only in-depth exploration that the authors undertake is the use of tele medicine. However even there, no new dimension or an argument is not emerging from the study.

Lack of support from higher ups and delayed referrals are not sufficiently analysed. Only authors would know if there is more analysis that can be brought in or not.

Quotations from providers are few and do not capture diversity of experiences.

The conclusion section definitely confirmed my overall observations -that there is very little connection between what was set ought, what was found and what was concluded. For instance, Telemedicine is being suggested for fulfilling “patient preferences and needs” while the entire paper has been on provider experiences! The need for training of GPs on effective health communication or the need for leadership skills which comes out as a finding in the previous section strangely does not find place in conclusion! More details need to be given for what exactly authors mean by a stronger team based support.

To summarize, the following are the major drawbacks of the paper according to me:

(a) Not identifying key research questions (beyond stating what are the experiences?)

(b) No insight or argument conveyed to readers, rather only describing participant experiences. All the ‘themes’ were more of a ‘tough and go’ discussion than a deep, evidence driven analysis

(c) Weak contextualization and conceptualization of the problem in focus.

Some minor queries/concerns:

• Were participants were informed a-priori that they would receive a 30GBP voucher for participation? Even if not, was it possible that others came to know of it (snowballing) which influenced self-selection?

• Line 67- please explain what is shielding

• Line 128- Participants were selected based on professional role, urban/rural- what were these roles? Was any comparison done based on roles or urban-rural divisions?

• Line 131- Was coding of transcripts done simultaneously as more interviews were being conducted? Or was done after all interviews were over? What kind of theoretical saturation was achieved?

Overall, the above observations do not convince me on the scholarly and program/policy level contribution of the paper, if selected for publication.

6. PLOS authors have the option to publish the peer review history of their article (what does this mean?). If published, this will include your full peer review and any attached files.

Reviewer #1: **Yes: **Sanjida Arora

Reviewer #2: No

---

## [Author Response · Author response to Decision Letter 0]

16 Nov 2022

Thank you for your feedback, we are very grateful. We have attached 2 documents - a new cover letter, as a more detailed address to the editors, and a Response to Reviewers document.

---

## [Decision Letter · Decision Letter 1]

8 Jan 2023

Primary Health Care during the COVID-19 pandemic: a qualitative exploration of the challenges and changes in practice experienced by GPs and GP trainees

PONE-D-21-40820R1

Dear Dr. Grut,

We’re pleased to inform you that your manuscript has been judged scientifically suitable for publication and will be formally accepted for publication once it meets all outstanding technical requirements.

Kind regards,

Alok Ranjan

Academic Editor

PLOS ONE

Additional Editor Comments (optional):

Reviewers' comments:

Reviewer's Responses to Questions

**Comments to the Author**

1. If the authors have adequately addressed your comments raised in a previous round of review and you feel that this manuscript is now acceptable for publication, you may indicate that here to bypass the “Comments to the Author” section, enter your conflict of interest statement in the “Confidential to Editor” section, and submit your "Accept" recommendation.

Reviewer #1: All comments have been addressed

2. Is the manuscript technically sound, and do the data support the conclusions?

Reviewer #1: Yes

3. Has the statistical analysis been performed appropriately and rigorously? 

Reviewer #1: N/A

4. Have the authors made all data underlying the findings in their manuscript fully available?

Reviewer #1: Yes

5. Is the manuscript presented in an intelligible fashion and written in standard English?

Reviewer #1: Yes

6. Review Comments to the Author

Reviewer #1: (No Response)

7. PLOS authors have the option to publish the peer review history of their article (what does this mean?). If published, this will include your full peer review and any attached files.

Reviewer #1: **Yes: **Sanjida Arora

---

## [Editor Report · Acceptance letter]

31 Jan 2023

PONE-D-21-40820R1 

Primary Health Care during the COVID-19 pandemic: a qualitative exploration of the challenges and changes in practice experienced by GPs and GP trainees 

Dear Dr. Grut:

I'm pleased to inform you that your manuscript has been deemed suitable for publication in PLOS ONE. Congratulations! Your manuscript is now with our production department. 

Kind regards, 

on behalf of

Dr. Alok Ranjan 

Academic Editor

PLOS ONE